# Research on the Non-Linear Relationship between Information Disclosure and Subsequent Purchases: The Moderating Effect of Membership Level

**Chunming Qin and Le Zuo ***

Department of Marketing, Economics and Management School, Wuhan University, Wuhan 430072, China; chmqin@whu.edu.cn
* Correspondence: zuole1010@whu.edu.cn

**Abstract:** In order to provide better marketing services to customers, companies often want to obtain as much customer information as possible. However, for customers, as well as leading to better service, information disclosure may also put them at risk of information leakage, so customers may respond in two different ways to firms' invitations to disclose information. This paper applies theories of social capital and communication boundary management to developing a framework for understanding the psychological mechanisms behind individuals' responses to disclosure invitations and their subsequent purchase behavior. The results of this study show that under the combined effect of social capital and communication boundaries, subsequent purchases show an inverted-U-shaped relationship, initially increasing and then decreasing as the level of disclosure increases. Furthermore, because the social capital of high-level members and firms is higher than that of low-level members, it moderates the inverted-U-shaped relationship; that is, the higher the level of membership, the more moderate the inverted-U-shaped relationship.

**Keywords:** information disclosure; social capital; communication boundary management; membership level; subsequent purchases

## 1. Introduction

In recent years, the rapid development of advanced technologies has considerably enhanced the collection and use of personal data, unlocking its diverse and convenient applications while maximizing its combined value. As a result, big data has emerged as a key production factor, especially in the modern retail industry. Companies are actively using huge amounts of customer data to gain in-depth insights into user preferences and effectively analyze them, ultimately striving to achieve higher marketing goals. The collection of comprehensive customer information has a direct impact on a company's economic value and competitive advantage. However, opinions on the collection of personal information vary. On the one hand, some companies strive to collect as much personal information as possible to inform their marketing campaigns. On the other hand, there are concerns that excessive data collection can lead to customer resentment. This concern stems from the frequent occurrence of data scandals, which have raised public concerns about the security of personal information [1]. While sharing personal information with a company can provide customers with better marketing services, there is also concern about the risk of information leakage. This dual aspect creates ambiguity in customers' attitudes towards disclosure.

Many countries have introduced broad information protection regimes that prohibit the collection of user data through illegal means. For example, in 2021 and 2022, China released the "Personal Information Protection Law of the People's Republic of China" and the "Implementation Rules for Personal Information Protection Certification", which impose strict restrictions on the collection of personal information by enterprises. Companies are

aware of the increased risk and difficulty of collecting information and the possibility that information collection practices may negatively affect the relationship between a company and its customers [2]. In this context, the collection of personal information by companies is increasingly dependent on the information disclosure behavior of individuals themselves and, to some extent, on consumers' willingness to share personal information [3]. As a result, the ensuing series of studies related to information disclosure has become a hot topic of interest.

In the existing literature, several studies have revealed the antecedents of information disclosure, arguing that information disclosure poses a risk of loss of personal information and therefore faces strong opposition from individuals [2]. This is due to the fact that the excessive collection of information raises strong information security concerns among individuals; for example, the use of information associated with relatively modern and unfamiliar technologies can increase the level of intrusiveness perceived by individuals and thus negatively affect consumer behavior [3]. However, it has also been argued that although people perceive information sharing to be risky, in practice, they often share their information freely [4]. People equate information sharing with feelings of connectedness and believe that disclosing information will not adversely affect their personal lives; they therefore disclose information publicly without thinking [5]. Kim argues that despite consumers' privacy concerns, they want the added value. He argues that people are less concerned about perceived privacy risks when providing personal information for better personalized services [6]. Poushneh et al. suggested that information interaction positively affects user satisfaction when users feel they have control over the company's use of their personal information [7]. It has also been suggested that when information is disclosed in a way that is characterized by immersion and engagement, consumers are less critical of information collection and more easily persuaded, which may translate into a higher willingness to disclose personal data [8]. On the other hand, recent studies suggest that feelings of comfort also influence customers' willingness to disclose information [2]. With regard to people's motivation to disclose, the prevailing research suggests that disclosure is the result of an individual's calculation of the risks and benefits of disclosure and that people will disclose information if they perceive the benefits to be greater than the risks [6,9].

Similar to the conflicting views on customers' willingness to disclose, there are two views regarding the effect of disclosure on subsequent behavior. The first view sees a positive effect of disclosure on subsequent behavior. Some scholars argue that information disclosure has an impact on the relationship between customer and company, promoting intimacy and subsequently influencing behavioral responses [10]. The act of disclosing information elicits a positive subsequent response when the individual confirms that he or she has control over the ownership of the information [11]. Recent studies have also found that individuals gain positive feelings when AR products are shown to them after disclosure [3]. The second viewpoint poses the exact opposite conclusion, arguing that information disclosure causes individuals to perceive an invasion of their privacy, which leads to avoidance behavior [12]. According to resistance theory, when consumers perceive a lack of control or loss of freedom, they try to resist persuasion and react negatively, and this negative effect spills over into brand attitudes and purchase intentions [3,13]. Overall, scholars have made some progress in the study of information disclosure, but existing studies have mostly examined the reasons for disclosure from the perspective of calculating actual benefits and risks, and there is still a lack of research on the psychological mechanisms through which individuals disclose information without clear benefits, and the effects of disclosure on subsequent behavior.

In fact, the disclosure of personal information serves as an important means for individuals to share information, and information sharing itself is a fundamental aspect of interpersonal relationships [14]. People satisfy their communication needs by disclosing an appropriate amount of personal information within a given social relationship [15]. Scholars argue that the amount of disclosure is related to the strength of the relationship. When individuals have a strong relationship with a firm, they may exhibit a relationship

enhancement bias. This bias leads them to interpret their disclosure behaviors in a way that assumes these actions are driven by positive intentions, thereby reinforcing positive behaviors and downplaying negative ones [11]. In this context, we aimed to investigate the impact of social capital and communication boundary management on individuals' disclosure behavior, particularly in situations where there are no explicit material gains. We combined social capital theory and communication boundary management to shed light on how social capital on both sides influences individuals' willingness to disclose personal information. In addition, we examined the subsequent effects of disclosure on individuals' purchasing behavior by examining the psychological changes that occur.

Our argument centers on the notion that individuals evaluate their communication boundaries based on the strength of a relationship, which is a reflection of their social capital. Higher social capital indicates stronger relationships and higher levels of trust between individuals and firms. As a result, individuals are more likely to share personal information within their communication boundaries because they perceive the firm as safe and trustworthy. In addition, sharing information deepens these relationships and has a positive impact on subsequent behavior. However, excessive requests for personal information can trigger negative psychological reactions, which in turn negatively affect subsequent behavior. In our study, we focused on information gathering in a department store as our research context and conducted field experiments to address the following issues: First, we aimed to clarify the non-linear relationship between information disclosure and purchase behavior by providing empirical evidence. Second, we explored the psychological mechanisms behind the impact of personal information disclosure on subsequent behavior, drawing on insights from social capital theory and communication boundary management. This approach allows us to investigate the reasons for the non-linear relationship between information disclosure and purchase behavior. Finally, we introduced the moderating effect of membership level to further validate the influence of social capital on disclosure. The theoretical value of this paper lies in filling the gaps in existing information disclosure research by investigating the psychological mechanisms of information disclosure and the relationship between disclosure and its influence on subsequent behavior. The study's results validate the theoretical derivation by showing a non-linear relationship between the degree of information disclosure and subsequent purchase behavior in a realistic department store scenario. They also confirm the influence of social capital through the analysis of real multi-level membership disclosure and purchase data. The study's practical value lies in helping companies to establish an understanding that information gathering is not entirely negative. In fact, moderate information disclosure helps to develop the relationship with customers and influences their subsequent behavior. This provides theoretical support for companies to better define information gathering and use information gathering strategies.

## 2. Literature Review

Personal information disclosure refers to the act of individuals sharing their personal information with another party [11,16]. Given the sensitive nature of personal information, the act of disclosure can make individuals feel vulnerable, thereby increasing their perception of intrusiveness [3,17]. Generally, information disclosure is perceived as risky, which makes people reluctant to share their personal information [18]. As technology has advanced and the capacity to collect and analyze personal information has increased, so has awareness of the potential risks associated with the disclosure of personal information [3]. However, despite these concerns, there is evidence that individuals are often less careful and even reckless when it comes to protecting their personal information [2,3,19]. It has been observed that individuals sometimes disclose their personal information without much thought [6,20]. Several studies have highlighted the dual nature of individuals' attitudes towards personal information, as they express concerns about the risks associated with disclosure, while at the same time showing a willingness to share their personal information [21,22]. This is because disclosure behavior is

related to an individual's perceived benefits, relationships with others, and perception of risk, which is based on context-specific trade-offs between expected benefits and perceived risks under specific circumstances and leads to decision-making behavior that integrates perceived risks and perceived returns [3,23]. For example, the most frequently cited disclosure benefit is gaining revenue, and the expectation of gaining revenue promotes disclosure behavior [18,24]. Acquisti et al. argue that disclosure decisions are not simply the result of a rational calculation of costs and benefits, but are also influenced by social norms, emotions, and inspirations, such as feelings of pleasure during the shopping process [14,18]. Moon et al. also argue that the relationship between two parties can have a significant impact on disclosure behavior, and that in the process of communicating, people pay attention to the impression they make on others [2,10]. In a recent study on the topic of information disclosure, Aboulnasr et al. argued that when consumers identify with a social networking site, they are more likely to feel that they will benefit more from sharing their personal information, at which point users are more likely to disclose their personal information via the social networking site [25]. This paper aims to analyze the influence of social capital on the disclosure and subsequent behavior of individuals, particularly in situations where there is no explicit price incentive or direct benefit associated with the service. We investigate the factors that impact individual information disclosure behavior when the return of disclosing information is primarily psychological rather than tangible. Moreover, we explore the psychological effects of information disclosure on the development of the specific relationship between the two parties, as well as the subsequent impact on purchase behavior.

Social capital encompasses the advantages that arise from social relationships and develop through a network of specific obligations and reciprocal expectations [26]. Social capital theory posits that individuals operate within a set of relational dynamics, wherein the outcomes of maintaining connections with others are influenced by factors such as goodwill, knowledge, influence, and trust. It is closely intertwined with the concepts of trust and relationships within social groups [27,28]. Social capital research primarily centers around the crucial role of relationships in providing social resources for action [29]. It emerges from the interplay of various factors, such as changing interpersonal dynamics, cultivating trust, fostering social networks, and establishing norms of reciprocity. Through these processes, social capital facilitates the creation of diverse networks of trust and encourages reciprocal actions between individuals or groups [28,30]. Nahapiet (1998) asserts that social capital, characterized by a strong foundation of trust within a business society, has the potential to reduce transaction costs between parties [26]. Similarly, Frenzen and Davis suggest that social capital accrues when one exchange partner extends a favor or gift to another partner [28].

Humans are inherently social beings, and the act of sharing information is a fundamental aspect of human interaction [14]. The ability to express thoughts and emotions with others holds significant value for individuals [31]. The desire for social interaction, recognition, and reputation serves as a motivational factor for people to engage in information disclosure [2,14,31]. Individuals perceive disclosure as a means of communicative sharing, which triggers neural mechanisms associated with reward processing [31,32]. Therefore, we argue that in situations where firms and customers have accumulated social capital, customers are likely to be inclined to respond to requests for information disclosure as a way to reciprocate the social capital they have received, even in the absence of explicit rewards. Information disclosure can also lead to positive action responses. Disclosure behavior is seen as one way of building intimacy, and disclosure behavior further strengthens the relationship [7,14]. Research by Moon further shows that in-depth disclosure builds a strong relationship between the parties, and this relationship will have a positive impact on subsequent behavior [2,3]. Research has also shown that not only do people tend to be attracted to the discloser, but the discloser is also attracted to the person to whom the information is disclosed [10,16]. Other studies have confirmed the same findings, indicating that as individuals engage in progressively deeper levels of disclosure, it frequently

results in the development of enduring, resilient long-term relationships characterized by mutual commitment and loyalty on both sides [11]. When committed relationships achieve this, it simultaneously fulfills the objectives of effective social communication and risk mitigation [3,33].

Furthermore, as the level of disclosure increases, it promotes the organic progression of the relationship from superficial to profound, thereby amplifying the power of persuasion within the dynamic [11]. A study by Aboulnasr et al. found that information disclosure positively impacted consumer brand engagement dimensions, including consumption, contribution, and the creation of brand-related content. Their research shows that consumers' willingness to share personal information on a social networking site facilitates the process of consuming, contributing to, and creating brand-related content [25]. This also serves as a reminder to the individual of the perceived strength of the relationship, influencing the individual's subsequent positive actions within the relationship, and ultimately positively impacting subsequent marketing activities [3,34]. Additionally, Kim et al. argue that information disclosure plays a crucial role in enhancing trust levels, and trust, in turn, facilitates consumer responses to marketing activities [35]. As the level of information disclosure increases, individuals reaffirm their trust-based relationship with the firm, resulting in positive responses to the firm's invitations and eventual direct purchases [6]. Building on this, we hypothesized that information disclosure fosters a close relationship between the firm and the customer whose information is shared, thus prompting the customer to respond positively to the firm regarding their subsequent purchasing behavior. And the intensity of the positive sentiment associated with the disclosure act intensifies as the level of disclosure deepens, thus amplifying the positive response to subsequent purchases.

## 3. Research Hypotheses

### 3.1. Information Disclosure and Communication Boundary Management

As mentioned above, social capital requires individuals to foster and sustain social relationships [28], and personal information sharing is an essential component in building significant long-term relationships [17], so information disclosure in social communication is inevitable. However, due to the vulnerability brought by information disclosure to individuals [3,10], the more information disclosure that is required, the stronger the risk awareness of individuals. There is no doubt that there is a tempering principle in the degree of disclosure.

According to communication boundary management theory, there exist information disclosure boundaries that are shaped by social norms and guidelines, governing the extent to which individuals are willing to share personal information with external parties and the general public [24]. As communication is central to communication boundary management, the nature of communicative interaction is intrinsically linked to relationships. Consequently, individuals often consider the level of closeness in their relationships as a reference point when deciding on the extent of their disclosure. This process helps establish an information exchange boundary, delineating the ownership and control of personal information [18,36]. Within this boundary, individuals perceive the recipient of the disclosed information as trustworthy, and they maintain a sense of control over the flow of information [10,11]. People establish these boundaries for various reasons, including the desire for intimacy and psychological relief, while also seeking to maintain independence from social influence and control [18,33].

In order to achieve the dual goals of maintaining social relationships and protecting personal information, individuals evaluate their relationship with the recipient of the disclosure and establish an appropriate boundary for information exchange [37]. When a customer receives an invitation from a company to disclose information, even if there is no clear immediate benefit, the customer may still be willing to disclose information within a certain communication boundary as a way of reciprocating and contributing social capital to the other party. Within this boundary, individuals form beliefs about the flow of perceived information, assess the trust and controllability of the relationship, determine

their own preferences for information sharing, and place trust in the default rules governing information flow [35]. When sharing information within this boundary, individuals assume that the recipient will adhere to the information management rules. They implicitly expect the other party to use and protect the disclosed information appropriately [11–36].

Based on earlier research, we believe that information disclosure can lead to positive customer action in relationships and positively influence subsequent behavior [11], but we suspect that this can only be achieved when the level of information disclosure is at a moderate level. Some studies have shown that excessive information collection triggers individuals' concerns about the risk of information loss, which leads to avoidance behavior [3,38]. Therefore, we hypothesize that when the level of individual disclosure remains within the communication boundary set by the individual, it will have a positive influence on the relationship and facilitate subsequent purchases. However, it is important to note that this effect may not be sustained indefinitely. If the level of disclosure exceeds the individual's communication boundary, the individual may perceive it as an over-collection of information, leading to a decrease in the willingness to disclose and a weakening of persuasion [35]. This, in turn, may result in resistance to marketing activities, including purchasing [3,39]. As the level of excessive disclosure continues to rise, avoidance behavior will be further exacerbated [2]. This is because when the level of information disclosure breaches the established sharing rules, it disrupts communication boundaries [18,36]. Individuals perceive a violation of information exchange norms and a decrease in personal control over the release and dissemination of information, which heightens their concerns about the risk of information loss [11,40]. In this case, when individuals feel that their personal information is being disclosed beyond their established communication boundaries, they take actions to regain control of their information. This not only affects their willingness to disclose information but also has a negative impact on their subsequent behaviors [3,10,38]. Therefore, we hypothesize that when the level of disclosure exceeds the communication boundaries set by the individual, it triggers negative psychological feelings that in turn negatively influence subsequent purchase behavior. Furthermore, as the level of disclosure increases further, the negative effect on subsequent purchase behavior becomes more pronounced. In light of the above analysis, we believe that there is a critical point at which information disclosure makes a positive contribution to subsequent purchase behavior, beyond which a continued increase in the level of information disclosure has a negative impact on purchase behavior. In other words, there is a non-linear inverted-U-shaped curve relationship between information disclosure and subsequent purchase behavior. Beyond this critical point, the level of information disclosure will have a diminishing effect on purchase behavior and may even have a negative impact.

This study further analyzes the deeper reasons for the above-mentioned inverted-U-shaped curve relationship. Firstly, this paper argues that the inverted-U-shaped relationship between information disclosure and subsequent purchase behavior is due to a combination of the positive effects of individuals disclosing information within the sharing boundary (disclosure reinforces the relationship) and the negative effects of disclosing information beyond the sharing boundary (too much information disclosure causes resistance). Individuals who establish information sharing boundaries based on the social capital they have accumulated with each other are actively engaged in maintaining the social relationship. This proactive behavior has a positive impact on their perception of the relationship, leading to corresponding positive effects on subsequent purchasing behavior. The level of information disclosure and the strength of the relationship between the parties directly influence the extent of the positive impact on subsequent purchases. Specifically, higher levels of information disclosure and stronger relationships tend to yield greater positive effects on subsequent purchase behavior. However, the positive effect of information disclosure on subsequent purchases does not always last; a critical point occurs when a certain level of disclosure is reached. As defined by communication boundary management theory, since individuals do not always perceive the presence of information risk at the time of disclosure, it is the salient cues that motivate individuals to focus on the risk of

disclosure, which leads to a negative response [3,14]. Therefore, when disclosure exceeds the sharing boundaries set by the individual, the individual is influenced by this salient cue: the "alarm bell" is sounded, the tipping point is met, and the individual is likely to respond negatively regarding their subsequent behavior [10,11]. If the level of disclosure continues to increase beyond this point, then persuasive behavior elicits even stronger resistance and the individual perceives that their personal freedom is being restricted, further exacerbating avoidance behavior [11]. Thus, the higher the level of disclosure within the communication boundary of information, the more positive the subsequent purchase response. Once the communication boundary of individuals' willingness to share information is exceeded, a turning point occurs where the effect of disclosure on purchase behavior shifts from positive to negative, and this effect strengthens as the level of disclosure continues to increase. In other words, as the level of information disclosure increases, subsequent purchases exhibit an inverted-U-shaped curve, initially increasing and then decreasing. Based on the above analysis, we propose the following Hypothesis 1.

**Hypothesis 1.** *The relationship between disclosure and subsequent purchase amount shows an inverted-U-shaped curve; that is, as the level of disclosure increases (low to moderate to high), the subsequent amount an individual purchases increases and then decreases.*

### 3.2. The Moderating Effect of Membership Level

Social capital is defined as a resource built in social relationships, accumulated in the form of a set of specific relationships and reciprocal expectations [30]. In the retail industry, customer management programs such as membership hierarchies and customer loyalty schemes are frequently implemented to foster and nurture these relationships [41,42]. Membership levels are usually set based on historical data such as purchase frequency, customer retention, and type of products purchased and are an indication of the level of social capital between the member and the company [43]. High-level members receive extra attention and rewards from companies [44,45]. Customers in a membership system have a relationship with the company built on past transactions, repeat purchases, etc. They will feel a stronger connection to the company than the average customer [42]. As members perceive themselves as a privileged group within the organisation, they will be more responsive to the organisation [42,46]. Of these, higher-level members have higher self-identity and perceived social status within the organization and stronger relationships with the firm than low-level members [47]. At the same time, members at different levels have access to exclusive resources that are not available to others, thanks to the social capital accumulated by both the individual and the service provider. As a result, they are more likely to respond to invitations from service providers. Moreover, the higher the social capital of both the individual and the service provider, the more positive the response [48].

Social capital emphasizes the individual's sense of responsibility or obligation based on the resources given to the individual by the organization [28,29]. In contrast, information disclosure is an important means of building social capital between parties, achieving the natural evolution of interpersonal relationships from superficial to intimate [5,36]. As previously discussed, the act of disclosure is a delicate balancing act between individual vulnerability and the dynamics of social relationships. When information is shared within the communication boundaries, it can be seen as an intentional effort to foster and maintain a positive relationship between the individual and the recipient of the disclosure [11]. In turn, the communication boundary between the individual and the subject of the disclosure is attributable to the closeness of the relationship, and the social capital accumulated by both parties may become a key determinant of the communication boundary, requiring appropriate attention to the relationship between the discloser and the subject of the disclosure [5,36]. It was also shown in a longitudinal study of disclosure behavior that people prefer to disclose more to familiar recipients and less to strangers [11,18,49]. From this, we believe that compared to low-level members, high-level members have higher social capital with companies, pay more attention to the maintenance with companies,

have higher levels of access to established communication boundaries, and are ultimately willing to disclose more information.

As trust and control are two core elements of communication boundaries, when individuals trust the person to whom they disclose, their concerns regarding information loss caused by personal information disclosure are alleviated [6,24]. Trust can develop from interpersonal relationships, and also from organizational relationships [5]. Pamela argues that membership affects an individual's trust in an organization and that a uniform interpretation of organizational rules by the membership hierarchy increases members' trust in the organization [50]. In contrast to low-level members, high-level members have higher social capital with the firm, stronger relationships with the firm, higher trust, and lower concern for information risk when disclosing information [2,19] and tend to respond reflexively to invitations to disclose information, that is, without extensive thought or deliberation [5,10]. This low perceived disclosure behavior has a weaker psychological impact, which also reduces the impact on subsequent behavior and has a lower positive impact on purchase behavior. Correspondingly, when the level of customers' information disclosure exceeds the boundary of information exchange, the high social capital of high-level members, compared to low-level members, will reduce the negative impact of the risk of loss of activation information [3,51], so that the negative impact on subsequent purchase behavior is smaller. That is, high-level members establish greater communication boundaries with firms than low-level members, and higher levels of disclosure have less impact on subsequent purchase behavior. As a result, the threshold of the inverted-U-shaped curve appears later, and the shape of the curve is flatter. Research Hypothesis 2 was thus proposed:

**Hypothesis 2.** *In contrast to low-level members, high-level members negatively moderate the inverted-U-shaped relationship between information disclosure and subsequent purchase behavior, resulting in a moderately inverted-U-shaped curve.*

According to the above assumptions, a comprehensive conceptual model of this study was proposed, as shown in Figure 1.

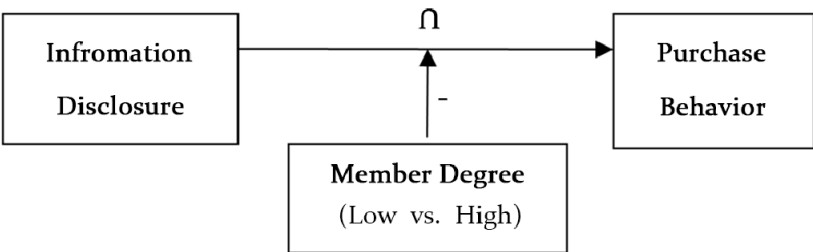

**Figure 1.** Conceptual model of the study.

## 4. Methodology

### 4.1. Data Collection

To validate the above model, we conducted a large-scale field experiment in cooperation with a large retail company in Southwest China (the company requested that its name be kept confidential). The company is one of the top three companies in the consumer sector in its city and incorporates a variety of business formats, including large shopping malls, supermarket chains, home appliance supermarkets, and online shopping malls, while selling a wide range of product types, including men's and women's fashion, shoes and bags, food, home appliances, and so on. By the end of 2018, the company had 1.2 million members and an impressive annual revenue of nearly RMB 2 billion. To facilitate effective customer management, the company has implemented a comprehensive customer relationship management system that meticulously tracks consumers' personal information and purchasing behavior. As a result, the company's data is a trusted and representative source within the retail industry. In keeping with the overriding principles of privacy

and security, the company takes measures to anonymize personal names and any other personally identifiable information during data provision.

In July 2019, we drew a sample of 130,000 from a membership pool of 1.2 million via random sampling. According to the definition of churned customers used in a study by Thomas et al., in the context of the firm, members who have not bought anything for 3 years are considered to have left the firm [52]. Therefore, we removed this part of the sample, leaving a final sample of 121,455. The enterprise sent a questionnaire (Table A1) to these 121,455 consumers via mobile phone SMS, inviting them to fill in a total of 48 items in the questionnaire. The collected data included 8 basic personal information questions (regarding age, gender, mobile phone number, and area of residence), 16 items related to consumption preferences (including types of products of interest, preferred shopping channels, and shopping recommendations), and 24 items regarding daily entertainment preferences (such as favorite foods, leisure activities, and sports). The questionnaire was administered over the course of one week. Following the completion of the questionnaire, we closed it and proceeded to send SMS messages to consumers via their mobile phones. The message read, "Dear members, from 8.1 to 8.4, you can enjoy special offers at xx Department Store! Get a 50% discount on home appliances and earn high points for members!" The campaign spanned one month, during which consumer purchase behavior data were observed and tracked. The final analysis was carried out by combining the data from the first three months of the trial.

*4.2. Measurement of Variables*

4.2.1. Purchase_after

The dependent variable in this study was the average weekly number of consumers who made a purchase one month after sending a marketing campaign text message, including the sum of purchase amounts for all businesses. Because of the significant right-skewed pattern of this variable, a log value was taken for the purchase amount:

$$Purchase\_after = log\ (average\ weekly\ purchase\ amount + 1)$$

4.2.2. Disclosure Degree

The independent variable in this study was the degree of information disclosure offered by consumers in the questionnaire, measured by the total number of completed questions [5]. We defined the completion of each question item in the questionnaire as 1 and non-completion as 0 and used the sum of these numbers to obtain a measurement of the degree of information disclosure.

4.2.3. Membership Level

The moderating variable in this study was membership level. The company's membership level system classifies members at different levels by calculating the sum of the customer's daily spending in all formats and the point value corresponding to the accumulated spending amount, and when the accumulated point value reaches a certain threshold, the customer's membership level is upgraded from the current membership level.

4.2.4. Control Variables

In this study, we decided to use consumers' purchase behavior prior to completing the disclosure questionnaire as control variables, including the average weekly purchase amount (Purchase_pre), the average weekly product variety (SKU_pre), and the average weekly number of purchases (Trips_pre) in the previous three months. These variables have an impact on consumer participation in the questionnaire, as the closer they are to the company, the more likely they are to participate [42,43]. They also affect subsequent purchase behavior after completing the questionnaire, i.e., Purchase_after, as consumers who are more closely connected to the company are more likely to make repeat purchases and spend more on average [46]. We therefore controlled for the three variables mentioned above.

In addition, this study controlled for age, gender, and other variables that would affect consumer purchase behavior in the literature [53,54]. To avoid the effect of outliers, all of the above variables were subjected to a 0.05 tail reduction treatment.

## 5. Results and Analysis

### 5.1. Descriptive Statistics and Correlation Analysis

In this study, descriptive statistics and correlation analysis of the variables were first carried out, and the test results are shown in Table A2. As can be seen from the table, the total sample size was 121,455, and the final sample size was determined to be 95,030 due to missing information regarding some membership levels. The linear relationship between disclosure level and purchase amount was positive (0.087, $p < 0.05$) and the linear relationship between membership level and purchase amount was positive (0.084, $p < 0.05$). Overall, there was no strong correlation between the variables, with the correlation coefficients all below 0.5. Next, a multicollinearity test was conducted on the matrix of variables, and the results showed that the mean value of VIF was 1.66, which was much smaller than the critical value of 10, so there was no collinearity between the variables.

### 5.2. Regression Analysis

In order to verify the non-linear relationship between the variables, the following analytical models were constructed and regressed in this paper, and the regression results are presented in Table A3, including the results of the three model regressions. Model 1 included only the linear relationship and control variables of the independent variable (Disclosure_degree), Model 2 added the quadratic term of the independent variable, and Model 3 added the moderating variable. The moderating relationship was verified for both the primary and quadratic terms.

$$\text{Purchase\_after} = \alpha + \beta_1 \text{ Disclosure\_degree} + \beta_2 \text{ Disclosure\_degree}^2 + \beta_3 \text{ Membership\_degree} + \beta_4 \text{Disclosure\_degree} \times \text{Membership\_degree} + \beta_5 \text{ Membership\_degree} \times \text{Disclosure\_degree}^2 + C_i + \varepsilon_i$$

According to the results of Model 2 in Table A3, an inverted-U-shaped effect of the level of disclosure on the purchase amount was found. The coefficient $\beta_1$ of the primary term (Disclosure_degree) was positive, the coefficient $\beta_2$ of the quadratic term (Disclosure_degree$^2$) was negative, and both were significant ($\beta_1 = 0.259$, $p < 0.01$; $\beta_2 = -0.006$, $p < 0.05$). According to the results of Model 3 shown in Table A3, it is possible to observe the moderating effect of the moderating variable Membership_degree on the inverted-U-shaped relationship between the independent variable (Disclosure_degree) and the dependent variable (Purchase_after) ($\beta_2 = -0.006$, $p < 0.05$; $\beta_5 = 0.005$, $p < 0.05$). The coefficient $\beta_2$ of the quadratic term of the independent variable (Disclosure_degree$^2$) was negative, the coefficient $\beta_5$ of the interaction term between the quadratic term of the independent variable (Disclosure_degree$^2$) and the moderating variable (membership_degree) was negative, and all of these are statistically significant. This indicates that the moderating variable Membership_degree has a negative moderating effect on the inverted-U-shaped relationship between disclosure degree and purchase amount, meaning that the higher the membership_degree, the more moderate the inverted-U-shaped relationship curve between disclosure and subsequent purchase amount. Thus, Hypothesis 2 was verified and supported.

### 5.3. Robustness Check

We conducted a robustness test to ensure that our main results were robust. Due to the missing information regarding membership levels, 26,425 samples were dropped, and the final sample size was determined to be 95,030. Rather than dropping the samples with missing values, we replaced the missing values by matching the closest sample by characteristic variables, including Purchase_pre, SKU_pre, and Trips_pre. In this way, the

final dataset has 121,455 observations, and we ran the regression analysis again with the new dataset. The results are robust, and the hypotheses are supported.

*5.4. Endogeneity*

This study presents a comprehensive model that considers the potential endogeneity of disclosure degree by incorporating relevant control variables. Nonetheless, we acknowledge the possibility of unobserved factors simultaneously influencing both disclosure degree and purchase behavior subsequent to disclosure. To mitigate the impact of unobservable omitted variables, we adopted a Gaussian copulas approach [55], a robust instrument-free method widely used in management and marketing research when valid instruments are challenging to identify [56,57]. Gaussian copulas enabled us to model the joint distribution between the error term and the endogenous variables through control function terms. These control function terms, when included in regressions, effectively account for the correlation between the error term and the endogenous variables. The findings, as demonstrated in Table A3 Model 4, consistently support our model, thereby mitigating potential endogeneity issues inherent in this study.

## 6. Conclusions and General Discussion

### 6.1. Conclusions

In the era of big data, companies actively encourage customers to disclose information; however, there is a research gap in the industry regarding the non-linear relationship between individual information disclosure and subsequent behavior. Based on social capital theory and communication boundary management theory, this study adopted the information gathering process commonly used in department store retailing as the research context. Using a questionnaire survey and a large-scale field experiment, the study examined the relationship between information disclosure and subsequent purchase behavior in the absence of substantial returns and investigated the underlying mechanisms. Three main conclusions emerge from the research.

First, information disclosure behavior is influenced by the social capital of both parties. Even if there is no substantial benefit return, individuals will still actively disclose personal information within a moderate level of information disclosure. At the same time, the positive effect of information disclosure behavior promotes the response of subsequent purchase behavior.

Second, the relationship between information disclosure and subsequent purchase behavior does not conform with the absolute linear relationship found in previous studies. This study found that, due to the presence of communication boundaries, as the level of personal information disclosure increases, the degree of disclosure and subsequent purchase show a non-linear inverted-U-shaped curve relationship. That is, when the level of information disclosure is low or high, subsequent purchases are lower. Subsequent purchases are highest when the level of information disclosure is moderate.

Third, this study finds that membership level moderates the inverted-U-shaped curve relationship between disclosure and subsequent purchase behavior. When individuals continue to disclose information beyond the communication boundary, the social capital of high-level members and firms moderates the perceived risk of disclosure and reduces the negative impact on purchase behavior compared to low-level members. In other words, compared with low-level members, high-level members have higher social capital and stronger relationship with companies, and information disclosure is more casual, so the positive perception of their relationship with the company brought by information disclosure is lower compared with low-level members, thus moderating the positive effect on purchasing behavior.

In addition, the effects of gender and age on marketing effectiveness have been generally validated in previous studies [53,54]. For example, women demonstrate higher participation in marketing activities, and older customers are less willing to participate in

activities due to social activity, consumption power, and other reasons. Thus, we controlled for gender and age variables in the experiment to ensure the robustness of the results.

### 6.2. Theoretical Contributions

These findings enrich the research on the antecedents of information disclosure and the mechanisms by which disclosure affects behavior. Specifically, we examined the causes and moderators of the inverted-U-shaped curve relationship between information disclosure and purchase behavior. Our theoretical contributions are mainly in the following areas. To begin with, most of the existing studies have explored the linear positive or negative effects of information disclosure on subsequent behavior from a single perspective, but in reality, such issues cannot be accounted for and explained by a linear relationship. This study proposed and tested a non-linear relationship between information disclosure and subsequent purchase behavior in a realistic retail department store scenario, providing a corresponding explanation for the inconsistency of existing research findings. Meanwhile, this study analyzed the psychological mechanism of action behind the inverted-U-shaped relationship deduced from the literature and this paper argues that social capital and information communication boundaries are the root causes of the inverted-U-shaped relationship between information disclosure and subsequent purchase behavior. Secondly, the present paper extends the application of communication boundary management to the marketing field. Starting from the psychology of information disclosure, we examined how individuals consider the boundary between disclosure and concealment when making disclosures, further explaining the reasons for the emergence of the inverted-U-shaped curve relationship. Finally, this study showed that the relationship between the discloser and the receiver moderates the subsequent purchase behavior. Therefore, we can conduct empirical studies on the moderating factors that may affect the relationship in different industries or different cultural contexts, thus continuously enriching the theoretical research on this topic.

### 6.3. Practical Contributions and Implications

The current study has practical implications for the information collection and subsequent marketing activities of enterprises.

Firstly, the collection of information is an indispensable part of a company, but the negative reactions caused by the collection of information is caused for concern. How data can be collected in a way that does not provoke negative reactions from customers to subsequent marketing activities is a real concern for companies. The findings of this study help to establish the view that information collection is not simply negative, but that certain information disclosure may also have a positive impact on the relationship between the company and the customer.

Secondly, the positive impact of information disclosure is limited, and it is moderate information sharing that strengthens the positive relationship between customers and a company. We need to focus on the principle of moderation in information disclosure and analyze the boundaries of information exchange that customers of different relationship strengths in terms of the social capital of the customer and the company. This means that high-level members with higher social capital are more willing to share more information. Thus, it is important to implement a differentiated information disclosure solicitation strategy for different membership levels.

Thirdly, membership level affects the inverted-U-shaped curve of the disclosure and purchase relationship. Therefore, in practice, enterprises should be sure to provide corresponding revenue strategies for different levels of customers through means such as membership hierarchy, continuously improving the social capital of enterprises and members, promoting their identification with enterprises, and expanding their information interchange boundary with enterprises, thus obtaining a more positive marketing response to enterprises.

Fourthly, research related to information disclosure often deals with specific economic or service benefits, but this paper's study of the positive contribution of moderate information disclosure to subsequent purchase behavior without substantive beneficial returns may provide new ideas for companies to experiment more with low-cost marketing when collecting information.

*6.4. Limitations and Future Research*

As a first exploration of and attempt to examine the impact of information disclosure on shopping behavior in department stores, this study has shortcomings. Perceptions of personal information are often associated with privacy, perceptions of privacy vary widely across populations [3], and information sensitivity varies across domains; for example, individuals are more sensitive regarding their medical or financial data than consumer information [6,19]. This paper did not explore whether the effect on subsequent behavior differs across these perspectives, and future studies could seek to further test whether the relationship between disclosure and purchase differs across different types of sensitive information from the perspective of information sensitivity.

Another limitation of our study is that different levels of membership data were used for validation, but no distinction was made between members and non-members. We assume that a firm's members should have higher social capital than the average customer, but due to the specific nature of the information, it is possible that non-member customers are willing to provide more information to the firm, and therefore, different characteristics regarding their purchase behavior may emerge, which can be explored in the future. In addition, this paper concludes that the positive effect of information disclosure on purchase behavior is highest when disclosure is moderate, but this moderate level is only a rough range, and further research could be conducted in the future to explore a more precise level of disclosure at the information exchange boundary for different relationship strengths.

Lastly, as the results of this study are based on data from a single corporate retail department store in China, this may be representative of the general phenomenon of retail department stores in China. But can this one study's conclusions apply to countries with different cultural backgrounds, or in e-commerce scenarios? This will need to be confirmed with richer data and research in the future.

**Author Contributions:** Conceptualization, C.Q.; methodology, C.Q. and L.Z.; software, L.Z.; validation, L.Z.; formal analysis, C.Q. and L.Z.; investigation, C.Q. and L.Z.; resources, C.Q.; data curation, L.Z.; writing—original draft preparation, C.Q.; writing—review and editing, C.Q.; visualization, L.Z.; supervision, C.Q.; project administration, C.Q. and L.Z. All authors have read and agreed to the published version of the manuscript.

**Funding:** This research received no external funding.

**Informed Consent Statement:** Informed consent was obtained from all subjects involved in the study.

**Data Availability Statement:** The data presented in this study are available on request from the corresponding author. The data is not publicly available because it involves real customer consumption information, which requires the consent of the experimental companies to access.

**Conflicts of Interest:** The authors declare no conflict of interest.

## Appendix A

**Table A1.** Information Disclosure Questionnaire.

| Item | Select/Fill in |
| --- | --- |
| (1) Gender | Male; Female |
| (2) Age | |
| (3) Education | Specialty; Undergraduate degree; Master's; Doctorate |
| (4) Occupation | |

**Table A1.** *Cont.*

| Item | Select/Fill in |
|---|---|
| (5) E-mail | ____________________________________ |
| (6) Tel | ____________________________________ |
| (7) Address | ____________________________________ |
| (8) Monthly consumption amount | ____________________________________ |
| (9) Categories of recent purchases | Clothing; Bags; Cosmetics; Home appliances; Books; Furniture |
| (10) Where do most often purchase items? | Online shop; Offline shop |
| (11) Do you seek advice from friends or family before shopping? | Yes; No |
| (12) How many times do you go shopping per month? | 1~3; 4~5; >5; Other |
| (13) As a consumer, what factor do you favor most often? | Whether the money is enough; The usefulness of goods; Commodity grade; Other |
| (14) Do you usually look at the items you buy? | Sales; Price; Evaluation; Purpose; Brand; Other |
| (15) What are your favorite makeup brands? | ____________________________________ |
| (16) What are your favorite casual snacks? | ____________________________________ |
| (17) What are your favorite brands of clothing? | ____________________________________ |
| (18) What are your favorite brands of appliances? | ____________________________________ |
| (19) Do you like to stay in or go out at the weekend? | Stay home; Go out |
| (20) What time do you normally go to bed at night? | ____________________________________ |
| (21) What is your favorite way to relax? | Shopping; Playing cards; Playing sports; Reading; Watching films; Travelling; Going to art exhibitions Other |
| (22) What is your favorite form of exercise? | Walking; Running; Ball games; Climbing; Swimming; Jumping rope; Other |
| (23) How many times a week do you exercise? | Once; Twice; Three times; More than three times; Occasionally |
| (24) Do you like to comment on social networking sites? | Often; Depends; Occasionally; Never |

**Table A2.** Descriptive statistics and correlation data of variables.

| Variable | Sample Size | Mean | Standard Deviation | (1) | (2) | (3) | (4) | (5) | (6) | (7) | (8) | (9) |
|---|---|---|---|---|---|---|---|---|---|---|---|---|
| Purchase_after | 121,455 | −3.728 | 5.202 | 1.000 | | | | | | | | |
| Disclosure_degree | 121,455 | 0.282 | 2.445 | 0.087 * | 1.000 | | | | | | | |
| Purchase_pre | 121,455 | −0.079 | 5.51 | 0.235 * | 0.019 * | 1.000 | | | | | | |
| SKU_pre | 121,455 | 0.444 | 0.994 | 0.328 * | 0.055 * | 0.390 * | 1.000 | | | | | |
| Trips_pre | 121,455 | 0.118 | 0.181 | 0.390 * | 0.076 * | 0.556 * | 0.834 * | 1.000 | | | | |
| Sex | 121,455 | 0.283 | 0.45 | −0.017 * | 0.005 | −0.025 * | −0.018 * | −0.019 * | 1.000 | | | |
| Age | 121,455 | 37.472 | 10.942 | 0.029 * | −0.029 * | 0.041 * | −0.003 | 0.001 | 0.081 * | 1.000 | | |
| Membership_points | 121,455 | 1000.255 | 408.391 | −0.045 * | 0.002 | −0.063 * | −0.044 * | −0.070 * | −0.005 | −0.019 * | 1.000 | |
| Membership_degree | 95,030 | 6.254 | 1.124 | 0.084 * | 0.021 * | 0.037 * | 0.071 * | 0.105 * | 0.010 * | 0.059 * | −0.060 * | 1.000 |

Note: "Puchase_after" and "Purchase_pre" use the log value; * indicates significant at the 5% level.

**Table A3.** Regression result analysis.

| | DV | | | |
|---|---|---|---|---|
| | **Purchase_After** | | | |
| | **(1)** | **(2)** | **(3)** | **(4)** |
| **IV** | **Model 1** | **Model 2** | **Model 3** | **Model 4** |
| Disclosure_degree | 0.126 *** | 0.259 *** | 1.023 *** | 1.256 *** |
| | (0.006) | (0.054) | (0.293) | (0.375) |
| Disclosure_degree$^2$ | | −0.006 ** | −0.035 *** | −0.049 *** |
| | | (0.002) | (0.013) | (0.019) |

**Table A3.** *Cont.*

| IV | DV Purchase_after | | | |
|---|---|---|---|---|
| | **(1)** | **(2)** | **(3)** | **(4)** |
| | **Model 1** | **Model 2** | **Model 3** | **Model 4** |
| Membership_degree | | | 0.247 *** | 0.246 *** |
| | | | (0.016) | (0.016) |
| Disclosure_degree × Membership_degree | | | −0.128 *** | −0.147 *** |
| | | | (0.045) | (0.055) |
| Disclosure_degree$^2$ × Membership_degree | | | 0.005 ** | 0.006 ** |
| | | | (0.002) | (0.003) |
| Purchase_pre | 0.027 *** | 0.027 *** | −0.164 *** | −0.165 *** |
| | (0.003) | (0.003) | (0.004) | (0.004) |
| SKU_pre | 0.096 *** | 0.096 *** | 0.112 *** | 0.115 *** |
| | (0.028) | (0.028) | (0.028) | (0.028) |
| Trips_pre | 10.130 *** | 10.123 *** | 10.160 *** | 10.207 *** |
| | (0.165) | (0.165) | (0.166) | (0.167) |
| Sex | −0.141 *** | −0.142 *** | −0.085 ** | −0.098 *** |
| | (0.030) | (0.030) | (0.038) | (0.038) |
| Age | 0.014 *** | 0.014 *** | 0.005 *** | 0.004 *** |
| | (0.001) | (0.001) | (0.002) | (0.002) |
| Membership_points | −0.000 *** | −0.000 *** | −0.000 *** | −0.000 *** |
| | (0.000) | (0.000) | (0.000) | (0.000) |
| Copula Disclosure_degree | | | | 3.926 |
| | | | | (3.802) |
| Constant | −5.259 *** | −5.258 *** | −5.565 *** | −14.235 * |
| | (0.061) | (0.061) | (0.125) | (8.408) |
| N | 121,455 | 121,455 | 95,030 | 94,561 |
| $R^2$ | 0.158 | 0.158 | 0.122 | 0.12 |

Note: In the table, () is the robust standard error; *, **, and *** are significant at the level of 10%, 5%, and 1%, respectively.

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
