# Peer review of "Research on the Non-Linear Relationship between Information Disclosure and Subsequent Purchases: The Moderating Effect of Membership Level"

_systems, doi:10.3390/systems11080398_

Round 1
Reviewer 1 Report
This research is analyzing an interesting topic but, in my view, some relevant aspects need further clarification:
Variables: the information about the questions included in the questionnaire is very limited: a table with some basic statistics for the variables could be added. Descriptive statistics are mentioned in the title of the Section 41.1 but only correlations are presented; for some variables like, for example, disclosure degree, the mean in the correlation table is quite low (0.282), so basically the proportion of the items fully answered was very low (were people more reluctant to complete personal information questions or they were equally reserved concerning their purchasing preferences too? It would be also good to see what was the proportion of the sample that answered all the items and their distribution by age and gender). In the same fashion, membership level (low, high) is based on some threshold, but this reference value is not given. In Table A1, membership level’s average value is 6+, so presumably this variable is rated on a scale 1-10 and 5 could be the reference level? Also, it would be good to have the proportion of the respondents classified as low- and high-level members. In Tables A1 and A2, for the variable sex it is not clear whether the reference level it refers to men or women.
Also, in the Results and discussion section, control variables, like gender and age, came out to be significant but they are not mentioned in the discussion of the results.
Data seemed to be collected online, perhaps also during the Covid-19 pandemic? If this the case, this might have affected the expected outcome of the field work as it could have affected respondents’ purchasing behaviour.
In section 4.1. it is stated: “The linear relationship between disclosure level and purchase amount was positive (0.087, p < 0.05) and the linear relationship between membership level and purchase amount was positive (0.084, p < 0.05)”; given the p-value in both cases, the relationships are also significant, right?
The authors are fitting the U-shape for the data but, it is not clear whether the quadratic regression (U-shape) is the best fit for the data: one way to look at would be to fit first the linear regression (e.g., Y=a+b1X) and then add a quadratic term (Y=a+b1X+b2X2) and test whether the increase in R2 is significant. I wonder whether this step was performed by the authors.
Minor issues:
- title of the section 5 "Gerneral" - should be "General); 5.4 "Furture" - should be "Future"
- section 5.4 last parragraph: "... store in China, although this may" - I think the word "although" should be removed from the statement.
Author Response
Dear reviewers,
thank you so much for your valuable comments, which definitely helped us improve our paper. We hope that our revision and reply has acceptably addressed your concerns. The specific revisions and responses are attached for your review. Thanks again.

Reviewer 2 Report
The paper deals with information disclosure using two theories to explain the phenomenon. Overall, I feel the article has merit, but I suggest some changes.
First, the psychological mechanisms are indeed important. However, it is hard to understand the exact motives with a single empirical study. A short qualitative phase would help to enrich the empirical package.
Second, the paper is full of acronyms. Please spell them out. Similarly, there are other linguistic issues. For instance, the plural of the word hypothesis is hypotheses.
Third, the psychological literature has recently shown the determinants of disclosure. Please integrate this discussion following: Aboulnasr, K., Tran, G. A., & Park, T. (2022). Personal information disclosure on social networking sites. Psychology & Marketing, 39(2), 294-308.
Fourth, figure 1 does not clarify the sign of the expected relationships and some variables are ambiguous (it is unclear from there how you manipulate the independent variable and what are the levels of the dependent variable).
As a minor point, note that the references are in a different format.
Proofreading is required
Author Response
Dear reviewers,
Thank you so much for your valuable comments, which definitely helped us improve our paper. We hope that our revision and reply has acceptably addressed your concerns. The specific revisions and responses are attached for your review. Thanks again.

Round 2
Reviewer 1 Report
The authors gave full consideration to reviewer's comments and provided satisfactory answers.
A very minor issue: line 200, "Aboulnasr" I believe it should be "Aboulnasr et al." , as per reference no. 25.
Author Response
Dear reviewer,
Your comments are correct and our revisions are as follows:
Comment 1. A very minor issue: line 200, "Aboulnasr" I believe it should be "Aboulnasr et al." , as per reference no. 25.
Response 1. Thank you for your advice on this matter. We have carefully checked the manuscript and the reference and have indeed found this problem. We have now revised "Aboulnasr" to "Aboulnasr et al." (See on p.5, line 220).
Again, thank you again for your careful and meticulous reading. We hope that our revision and reply has acceptably addressed your concerns.
